# Design of Spurious Dynamic Inverter-Based Level Shifter with Error Tolerance for Robotic Arm Controller

**DOI:** 10.3390/mi15121431

**Published:** 2024-11-28

**Authors:** S. Vijayakumar, Lachi Reddy Poreddy, Mohammed Mahaboob Basha, Karnam Gopi, Srinivasulu Gundala, Javed Syed

**Affiliations:** 1Department of ECE, Sreenivasa Institute of Technology and Management Studies (Autonomous), Chittoor 517127, Andhra Pradesh, India; vijayakumar@sitams.org; 2Department of ECE, Lakireddy Bali Reddy College of Engineering, Mylavaram 521230, Andhra Pradesh, India; plreddy@lbrce.ac.in (L.R.P.); srinivasulugundala46@lbrce.ac.in (S.G.); 3Department of ECE, Sreenidhi Institute of Science and Technology (Autonomous), Hyderabad 501301, Telangana, India; 4Department of ECE, Vijayam Institute of Technology, Chittoor 517001, Andhra Pradesh, India; gopi.vijayamit@gmail.com; 5Department of Mechanical Engineering, College of Engineering, King Khalid University, Abha 61421, Saudi Arabia

**Keywords:** DIBLS, wide range, above-threshold, near-threshold, LS, sub-threshold

## Abstract

In robotic arm controllers, the ability to shift signal levels is crucial for interfacing between different voltage domains in a processor. The level shifter (LS) has been used to convert signals operating near threshold voltage to signals operating well above the threshold voltage. Researchers have developed current mirror-based LSs to employ current mirrors, which duplicate the current from one transistor and accurately replicate it in another, ensuring precise current matching. In this research, a dynamic inverter-based level shifter (DIBLS) with an error correction circuit is implemented. One of the main issues addressed by DIBLS is the problem of current disagreement. Current disagreement arises when multiple circuit components attempt to draw current from a common source, which leads to operational problems. Furthermore, DIBLS includes a feedback inverter controlled by the output node; this feedback inverter likely plays a pivotal role in controlling and stabilizing the output voltage of operation of the LS. The results demonstrate that DIBLS offers notable advantages on increased operational speed. This speed improvement has been achieved by circumventing the threshold voltage drop associated with the feedback of the inverter and by ensuring complete output swing, addressing stability issues. Voltage shifting between 0.3 V and 1.2 V at 1 MHz having power consumption is 16.57 nW, delay 0.22 ns, and energy per transition 32.25 fJ. The entire process is executed in 45 nm in CMOS technology.

## 1. Introduction

In the growing era of robotics, the demand for low power from battery systems such as mobile devices, IoT devices, and wireless sensors has been increasing tremendously. For the extension of battery life, there is a need for low-power designed circuits. In systems functioning with different blocks with different supply voltages, interconnecting with sub-blocks necessitates shifting the voltage level of the signals. This can be achieved by using voltage LSs that can shift low logic levels and sub-threshold levels to accurately acceptable voltage levels for the subsequent block. Nevertheless, sub-threshold functioning is one of the finest ways for achieving ultra-low power systems. Contrastingly, there is a disadvantage in the sub-threshold level due to deprived performance. To overcome this multi-supply voltage, it is used to maintain performance and power usage [1].

In electronic systems, multiple supply voltages (VDDs) are used to maximize energy efficiency and power usage. While lowering VDD can successfully cut down on both static and dynamic power consumption, it can also cause problems like possible instability and slower circuit operation. Many electronic systems use a multi-VDD structure to address these issues. In this structure, non-critical components use a lower supply voltage (VDDL) to maximize energy efficiency, while timing critical components operate using an elevated supply voltage (VDDH). It is crucial to make sure that signals in various VDD domains are not directly coupled in such multi-supply systems. As an alternative, a dependable voltage level converter called an LS is employed. The voltage at which a transistor switches on, or when it is around the threshold VDD, should be able to be converted by the LS to a higher super-threshold VDD. Two widely used LS designs are mentioned in the passage: the cross-coupled PMOS LS (CPLS), depicted in Figure 1, and the current mirror LS (CMLS), depicted in Figure 2. The pull-up latch powered by VDDH is turned on, which restricts the voltage conversion range that CPLS can achieve. It does have an issue when the input voltage (IN) is high, as there can be a substantial static short-circuit current, resulting in energy waste. However, according to research works [2,3,4,5], it is more efficient to decrease VDD near Vth in some applications.

DIBLS was developed using 45 nm technology and simulated for delay and energy corners. Section 2 outlines the state-of-the-art LS, Section 3 describes the proposed dynamic inverter-based level shifter, which is evaluated based on the simulation and measurement results of Section 4, and conclusions are described in Section 5.

## 2. The Literature Review on Advanced LSs

In this section, we present a concise overview of advanced LSs from the available literature based on circuit topology, technology node, and performance metrics to assess the best design and extend the work towards an efficient LS.

Sub-threshold circuits in [6] have generated a lot of attention for recent ultra-low power researchers, as cascaded CPLS offers a highly efficient sub-threshold microprocessor intended for sensor applications. It is refined at several design phases, such as ISA specification, micro-architecture assessment, and circuit implementation refinement. For that, micro-architectural choices in the sub-threshold regime are very different from those in the traditional super-threshold mode. An innovative architecture designed for general purpose sensor processing, the sub-threshold operation is called to display an ideal energy node on the circuit side. Nonetheless, the propagation delay becomes increasingly sensitive to variations in the manufacturing process, potentially diminishing the gains in energy scaling. This study conducts a comprehensive analysis of the influence of voltage and frequency on energy efficiency within a framework. By carefully selecting and implementing library cells, it represents a sufficiently low voltage for minimal voltage operation. Experimental measurements on silicon demonstrate that the subliminal processor achieves an energy of 2.6 pJ per instruction at a voltage of 360 mV and a frequency of 830 KHz. Furthermore, this study examines variations in frequency and minimal voltage in different scenarios to validate the adaptive adjustment of clock frequency and minimal voltage for optimizing energy efficiency.

A quick and energy-efficient LS that can change from input signals from sub-threshold levels up to the normal supply voltage is presented as CPLS with a diode nFET header, as in [7]. The converter’s strong performance is demonstrated by experimental data taken from the 130 nm check on the chip, which enables a smooth convert from 188 mV to 1.2 V without the requirement for intermediate supply levels. In order to accomplish this robustness, the converter combines a number of circuit approaches that are capable of handling the large changes in current characteristics that are commonly seen in circuits that are below the threshold. Moreover, a level converter can elevate the voltage of an input signal from other voltages within this designated range to enable dynamic voltage margining, finally bringing it to a constant 1.2 V.

The LS in [8], intended for an effective logic level of voltage translation in the realm of sub-threshold voltage levels and conventional elevated voltage levels, is named as a CPLS with a diode pFET insertion. Two PMOS diodes are used in this LS arrangement, which allows it to interface with sub-threshold logic inputs with stability. The suggested LS is especially useful for power conscious systems because it can adjust its switching delay in response to changing input logic voltages. Moreover, the LS has a more straightforward circuit design, which results in less energy usage compared with current sub-threshold LS designs. Our calculations offer empirical proof of the increased energy economy and greater performance that this new LS achieves, using a 0.18 mm CMOS technique to construct a test chip to validate its functionality. Our experimental results confirm that the published LS can reliably and functionally operate with input voltages as minimum as 127 mV while delivering a result in voltage of 1.8 V.

In the brief communication of [9], the LS named as an RCC pull-up LS introduces a voltage LS that operates at high speed and very low power consumption. Adopting a unique RCC pull-up network is the fundamental innovation, as it achieves a large reduction in dynamic power usage while also improving switching speed. This LS demonstrates the capacity to raise input signals to a superior ostensible supply voltage stage by performing voltage level conversion on signals that have voltage levels well lower than the threshold voltage of a traditional MOSFET device. Because of its simple design with few component parts, the suggested LS has a small silicon footprint. Moreover, it functions in an extremely energy efficient mode, making it extremely adaptable for implementation in utilizations with strict power limitations. Post-layout simulation results, conducted utilizing standard 180 nm CMOS technology, affirm the effectiveness of the presented circuit in successfully converting voltage levels from 80 mV. Notably, when operating under conditions at supply voltages ranging from 0.40 V to 1.80 V, and with an n frequency of 1 MHz, the LS exhibits a power dissipation of 123.1 nW and a propagation delay of 23.7 ns.

In [10], a novel LS named as the modified revised Wilson current mirror (MRWCM) LS, is abbreviated as an LS with current limiter. The purpose of this technology is to effectively change the sub-threshold voltage to the supply voltage while minimizing power leakage. With the growing complexity and speed of modern electronic designs, power consumption has significantly increased. To address this, system-on-chip (SoC) configurations are often designed, with some modules powered by a sub-threshold voltage to achieve a lower power consumption, while other modules operate with a voltage supply above the threshold. The LS is a critical component in this setup, as it facilitates the transition between sub-threshold and supply voltage levels while aiming to minimize power leakage.

To describe the voltage LS architecture for power efficiency, the LS with adaptive current limiter described in [11] is specifically designed to transform extremely low input voltage levels into larger voltage levels. Our suggested design includes a current generator and selectively engages in the course of transition periods when the input signal’s logic level differs from the intended output at the changes in variation of input to output signals with the base of voltage conversion, to the output of logic level to lower the dissipation of static power. Furthermore, we incorporate a mechanism that diminishes the pull-up apparatus’s strength when the pull-down apparatus’s strength is actively lowering the result node. By doing this, the circuit is guaranteed to function even in situations where the input voltage drops below a PMOS threshold value. We have carried out an extensive analytical study to examine how this suggested signal does not align with the desired output logic level.

The multi-supply voltage technique has emerged as a major concept in [12] (modern SoC architecture named as MSVD LS)for optimizing the balance between processing energy and speed efficiency. The function of LSs is essential for enabling smooth communication between various voltage domains. In this brief research note, we present a novel linear switching architecture designed to offer both large-scale and quick voltage conversion. The suggested circuit uses the multi-threshold CMOS method and has a distinctive architectural approach. Strong voltage shifting is guaranteed over a wide range, including the change over from the voltage domains below the threshold to those above it. Most importantly, it does it with minimal energy use and quick response times. In the context of a 90 nm technology node while accounting for variations related to process of voltage and temperature, our design consistently achieves reliable voltage conversion, effectively transforming 0.1 V input into 1 V output signals. Results from the simulations of efficiency of this novel LS are evidenced by the delay of 16.60 ns, static power as low as 8.70 nW, and an overall energy consumption of a mere 77.00 fJ per transition when subjected to a 0.20 V 1 MHz input pulse signal.

The power efficient LS progressed in [13], pushing feature sizes below the 30 nm threshold, as the critical concern in modern microprocessor design has shifted towards power density management. To address this challenge, various sophisticated techniques have been developed, including dynamic voltage scaling, which operates at voltage levels close to the threshold, and the incorporation of multiple voltage domains. These strategies have become imperative for mitigating power consumption, both static and dynamic. A pivotal element in the implementation of these methods is the LS, responsible for facilitating voltage translation between distinct voltage domains. An efficient LS must prioritize rapid speed operation and energy efficiency. In this research, a novel LS has the capability of effectively translating voltages spanning a range from 250 to 790 mV. Notably, this proposed LS demonstrates a 42% reduction in propagation delay, a 45% decrease in energy consumption, and a substantial 48% reduction in static power dissipation when compared with existing designs. Moreover, this LS displays symmetrical transition periods between rising and falling, even under extreme conditions, with a maximum variation in timing of up to 12% observed across the entire voltage range.

A novel positive low-to-high voltage LS may be used to adjust voltage levels across a broad supply range. The voltages that the suggested LS can withstand are up to three times the MOS transistor’s individual safe operating limit. The elimination of threshold voltage dependence and floating nodes, which are limitations of traditional high-voltage LSs, ensures different conversion ranges with appropriate driving capabilities. Multi-Vt technology at 45 nm–110 nm is used to construct the circuit, which converts 1.2 V input signals to output signals with voltages between the range of 1.2 V and 1.8 V. The suggested LSs’ switching power consumption is in the order of 50 nW, while its static power is 5–8 nW for each of its several outputs, according to post-layout simulation. A circuit known as LS helps circuits using various supply voltages communicate by converting the signal level between various voltages. There are CMOS inverters that level down LSs, which are typically sufficient to level down from high voltage to low voltage. On the other hand, there are a number of issues in the LS architecture, such as short circuit establishment and signal integrity while transforming from low-voltage signals to high-voltage signals [14,15,16,17,18]. Furthermore, the difficulty of managing multiple supply systems rises when attempting to reduce level conversion costs between various voltage regions devoid of compromising the design’s resilience and durability. The methods that essentially address processing voltages higher than the MOS transistor’s safe operating limits, like supply high-voltage transistors with a greater voltage at which they operate, limits using the same technology for processes, necessitating more area and masks and making the process more complex, employing circuit techniques that guarantee the highest voltage throughout, facilitating the design of a circuit that is a conventional CMOS process, and, in the final, stage lowering the delay without compromising the overall performance and dependability [19,20].

## 3. Proposed Dynamic Inverter-Based Level Shifter

The schematic diagram of the dynamic inverter-based LS (DIBLS) with integrated logic errors correction is depicted in Figure 3. The DIBLS design primarily consists of two main components: the error detection module and the level shifting module within the circuit.

The operational principles of the proposed DIBLS are elucidated in Figure 4 and Figure 5. Specifically, the operations during a transition from VDDL to VDDH and transition from VDDH to VDDL are described based on input voltage IN. This representation provides a detailed understanding of DIBLS’s behavior under varying input conditions.

Current Equations (1) and (2) represent the currents of P5 and N3 when operating in a sub-threshold region while at a high VDDH.
(1)$$I_{sub}=I_{0}\;e^{\frac{\left(V_{GSP5}-V_{T}\right)}{ɳV_{th}}}$$
(2)$$I_{0}=\mu_{0}\;C_{ox}\frac{W}{L}\left(n-1\right)V_{thN3}^{2}$$
where *V_GS_* is the potential across gate source, *V_T_* is the threshold voltage, *V_th_* is the thermal voltage, *µ*_0_ is the electron zero bias mobility, *n* is the threshold slope factor of MOS devices (1 + *C_dep_*/*C_ox_*), *C_ox_* is the oxide capacitance of MOS devices, *C_dep_* is the depletion capacitance of MOS devices, *W* is the width, and *L* is the length of channel.

Current limiting is the exercise in digital circuits of enforcing a higher restriction that can be added to a load to guard the LS, producing or transmitting glitch currents from signal transition consequences because of a short circuit or comparable trouble within the load based on Equation (3).
(3)$$I_{P1leakage}=\frac{W_{P1}}{W_{0P1}}I_{o}\;e^{\frac{\left(V_{gsP1}-V_{tp}\right)}{{ɳV}_{th}}}$$

### 3.1. Transition from VDDL to VDDH

In the circuit depicted in Figure 4, the multiplexer operates at a low input voltage level (IN) denoted as VDDL. When IN is at 0.3 V, it consistently maintains logic “1” during the transition from VDDL to VDDH. This is achieved by grounding the node VD and the transistors MN3 and MP3 at VSS. Consequently, the voltage across MN3 through MP3 gradually increases. The difference in threshold voltage between MP3 and VVDD is denoted as VDDH-|V_th_, MP3|. As this voltage difference increases, the driving strength amplifies, approximating VDDH. Utilizing the inverter, the contention between MP3 and MN3 is notably reduced. As a result, MN3 can effectively elevate the voltage at node VD without significant interference from MP3. Eventually, OUT transitions to a high state, followed by the charging of node VD. In conventional setups, there exists substantial current contention between the MUX and the LS connected to the virtual VDD.

### 3.2. Transition from VDDH to VDDL

During the transition from VDDH to VDDL, as depicted in Figure 5, IN is denoted as VDDH. When IN is at 1.2 V, it consistently maintains logic “1” during the transition from VDDH to VDDL. While the multiplexer operates at a high-voltage level, throughout this transition, both the node VD and the transistors MN3 and MP3 are grounded at VSS. Consequently, the voltage across MN3 through MP3 gradually decreases. The difference in threshold voltage between MP3 and VVDD is represented by VDDL-|V_th_, MP3|, with the driving strength gradually diminishing, resembling VDDL. By utilizing the inverter to alleviate contention between MP3 and MN3, significant suppression is achieved. As a result, MN3 can effortlessly lower the voltage at node VD with minimal interference from MP3. This culminates in the final transition of OUT to a high state, followed by the discharge of node VD. In the conventional setup, involving MUX and LS connected to virtual VDD, the current contention is minimal.

The proposed DIBLS introduces a novel approach to managing OUT transitions, departing from conventional LS designs that rely on OUT feedback or logic error correction circuits. Typically, OUT is incorporated within a positive feedback loop, resulting in slowed OUT transitions. For instance, when OUT is not a stabled VDDL or VDDH, the deactivation of MP3 and subsequent decrease in voltage cause an increase in VD, leading to the deactivation of MP2 and a stable OUT as VDDL or VDDH. This process often results in incomplete OUT swings or floating nodes, thereby compromising LS stability, energy efficiency, and speed. In contrast, the proposed DIBLS employs a unique approach with positive feedback. In this design, the rise of OUT triggers its acceleration by deactivating MP3 and further lowering the VD node. Similarly, during the falling transition, the decrease in OUT accelerates its descent by activating MP3 and charging up the VD node to deactivate MP2. This distinct positive feedback mechanism of DIBLS ensures that OUT remains liberated from suspended states, slope levels deprivation, and incomplete swings. The response of DIBLS is illustrated in Figure 6.

Moreover, this design characteristic also contributes to the superior speed of the proposed DIBLS compared with previous LS designs. Further elaboration on this advantage is provided in Table 1. The proposed DIBLS addresses five issues identified in previous LS designs, such as elimination of the necessity for employing current mirrors and cross-coupled PMOS, avoidance of the need to counteract pull-up latch effects through the utilization of an inverter, activation of full swing for OUT by connecting it to MN4-MP4, execution of level shifting through a single stage, and ease of logic error correction. 

The delay and power characteristics are evaluated at the most critical temperature corner, as depicted in Figure 7. The delay between falling and rising transitions is compared, considering both local and global process variations. The design temperature corner exhibiting the worst case scenario varies depending on the voltage and power conditions. Regarding the delay, the LS performance deteriorates at 40 °C due to the exacerbated power conditions, as device performance weakens at this temperature while power becomes stronger. Conversely, under high-power conditions, the temperature corner shifts to 120 °C, as the transistor drivability diminishes at elevated temperatures, impacting both delay and power. The delay of the proposed DIBLS varies across different temperature conditions, peaking at its strongest at −40 °C, while remaining lower in other temperature ranges. Conversely, for power comparison, it is weakest at −40 °C and progressively improves across temperature ranges, culminating in its strongest performance at 120 °C. These findings represent the optimal balance achieved in delay and power characteristics across temperature variations compared with previous circuit designs.

## 4. Performance Analysis of LSs

In this section, a comparative analysis is conducted between the proposed DIBLS and previous LS designs in terms of frequency or speed, energy per transition, and area usage. Post-layout simulations are carried out using 45 nm and interconnected models, which have been validated by silicon results. We discuss a comparative analysis of various LSs based on overcoming VDDH-powered latch limitations; freedom from static current, evaluating the absence of static current, which denotes the continuous flow of electric current through a circuit when it is not actively transitioning between states, achieving full swing output, determining the LS’s ability to provide an output signal with a complete voltage swing; immunity to floating nodes, ensuring that the LS does not contain floating nodes, which can be susceptible to dynamic noise and instability, thus safeguarding it against such disruptive influences; no need to generate supplementary voltage supply, investigating whether the LS can function without necessitating the generation of a supplementary voltage supply; delay, the crucial path in terms of propagation delay, often determining the maximum clock frequency and performance of the entire circuit leakage power; and energy per transition, energy per transition is a critical factor in dynamic power consumption.

### 4.1. Performance Analysis on LS Delay

In Figure 8, the delay performance of the proposed DIBLS is compared with state-of-the-art LS configurations. These comparisons are conducted under different voltage conditions, where VDDL is varied while VDDH remains constant at 1.2 V. The node values in Figure 8 represent the regularized delay metrics for each LS. Normalized delay is computed as the proportion of the delay exhibited by LS to that of DIBLS. This metric serves as a means to evaluate the relative temporal characteristics of the different LS configurations. The analysis presented in Figure 9 reveals patterns and values that are not entirely consistent with those observed in Figure 8, indicating disparities in the comparative performance of the LS configurations across varying voltage conditions.

The delay encountered in cascaded CPLS architectures is significantly higher, approximately two to three times greater, compared with the delay exhibited by DIBLS. This discrepancy arises due to the inclusion of multiple sequential stages within the CPLS configurations. Notably, CPLS configurations featuring a diode header demonstrate the poorest speed performance across a wide spectrum of VDDL values. This reduced speed can be primarily attributed to the adoption of dual-stage architectures, leading to pronounced issues related to current contention within these structures.

When the voltage difference between VDDL and VDDH is too large, such as in the case of VDDL = 0.3 V and VDDH = 1.2 V, the voltage level shifting to give up power is not achieved and therefore is not depicted in Figure 9. In the current reducer configuration, the performance is primarily influenced by the falling transition of the output signal OUT. It is crucial to maintain an adequate gate voltage for MP2 to mitigate excessive current contention between MP5 and MN2, ensuring full activation of MP1. However, this process may limit the speed at which OUT can be raised, resulting in a delay approximately two to three times longer in comparison with DIBLS. In contrast, the adaptive LS configuration allows only the gate level of MP3 to be fully reduced. This simplifies the activation of the node by lowering the gate level of MP3 close to ground. Consequently, the speed of the LS can be enhanced by approximately 1.5 to 2 times in comparison with the current-limiter LS. Still, appropriate to the presence of the stacked PMOS path involved in raising OUT, the declining delay of the LS remains approximately 1.2 to 1.6 times longer than that of DIBLS.

The MSVD LS design demonstrates a slower operational performance relative to the proposed DIBLS, with a delay ranging from 1.2 to 1.8 times longer. This delay stems from the use of three stacked MUX to facilitate the pull-up of the OUT signal. Similarly, the RCC pull-up LS configuration showcases approximately 1.1 to 1.6 times more minimal delay than the proposed DIBLS. This slower operational speed arises due to the requirement of raising the MN3_MP3 inverter utilizing the connected MUX.

### 4.2. Performance Analysis on LS Energy

The CPLS diode insertion structure demonstrates an operational speed comparable to the proposed DIBLS, owing to its single-stage conversion and ability to achieve full-voltage swing operation. However, under conditions of significant voltage difference between VDDL and VDDH, the speed of the CPLS with the diode insertion experiences notable degradation, resulting in a roughly 20% decrease in speed compared with DIBLS. This decline in speed stems from the inability of the CPLS with the diode insertion to promptly lower the OUT signal when VDDL and VDDH is large, as the source of MP2 must be adequately excited by the stacked PMOS before this can occur. The delay in gate charging for MP2 contributes to the observed reduction in operational speed relative to DIBLS. In the diode header and cascaded CPLS configurations, the pull-down devices are controlled by VDDL. Consequently, these designs exhibit significantly higher energy consumption, approximately 1.70–2.60 or 2.60–4.70 times greater compared with DIBLS.

Figure 10 presents the energy comparison between the proposed DIBLS and various advanced LS configurations. This comparison is conducted with VDDL fixed at 0.3 V, while VDDH is varied. Similarly, the energy comparison of the proposed DIBLS with different LSs is made. In these comparisons, the RCC pull-up configuration exhibits the slowest energy due to the degradation of the OUT slope. Across most LS setups, the energy remains consistent regardless of changes in VDDH, attributed to the opposing effects it imposes on LS performance. However, for configurations designed for low-power operation, variations in VDDH may impact the energy, as evidenced in Figure 11. However, specific factors differently affect their energy consumption.

The adaptive limiter LS exhibits somewhat higher energy consumption, approximately 1.0–1.2 times greater than DIBLS. This increase can be attributed to the presence of a direct path in the logic error correction component, specifically MN3_MP3 for the falling transition or MN4_MP4 for the mounting transition, which persists while awaiting the transition to complete. While a like path is present in DIBLS, the energy dissipation by the current is higher in the adaptive limiter LS due to its longer delay. Additionally, the logic error correction circuit operates only during the falling transition in DIBLS, whereas it functions during falling and rising transitions in the adaptive current limiter LS.

### 4.3. Performance Analysis on LS Power Delay

The analysis for the suggested DIBLS circuit is displayed in Figure 12 based on the power, delay, and VDDL parameters for various frequencies. The delay of 0.7 V is suddenly raised by different frequencies. At 3 MHz, the delay is decreased compared with other frequencies. The power at 0.3 V to all other voltages is continuously decreased until 0.8 V is constant. At 1 MHz, the power is decreased compared with all other frequencies. The proposed DIBLS with different frequencies power and delay are better than previous LSs. The investigation demonstrates that, when VDDL is increased, the power decreases and the delay increases.

The analysis for the suggested DIBLS circuit is displayed in Figure 13 based on the power, delay, and VDDH parameters for various frequencies. The delay at 0.7 V is continuously decreased for different voltages of VDDH. At 1, 2, and 3 MHz, out of all other frequencies, the delay is decreased. The power at 0.7 V is continuously increased to 1.2 V. The power at 2 MHz frequency is smaller than all other frequencies. The proposed DIBLS with different frequencies, power, and delay are better than previous LSs. The investigation demonstrates that, when VDDH is increased, the power increases and the delay decreases. As a result, the proposed DIBLS achieves an energy efficient characteristic that is quantitatively illustrated.

Comparison Table 1 depicts that the suggested DIBLS has the minimum area of 5.84 µm^2^ up-convertible VDDL from 0.3 V to 1.2 V and is the strongest among the contrast circuits, with a lower area of around 75% of LS shown in [21]. It has a delay of 0.22 ns, which is negligible compared with other LSs of comparison. The power consumption of DIBLS is 16.57 nW, which is approximately just 1 nW higher than the best LS [25] and comparable to other LSs [22,24,25]. The simulation results in use are at different process corners, which can be suggestive of all possible technology variations; at the same time, the suggested DIBLS outperform the other circuits despite the inverter-type design limitation making the results even more significant. The proposed DIBLS is strong not only in terms of an ample translation range but also efficient in terms of power and delay. It is a design only capable of performing up-shift or down-shift.

### 4.4. Area Comparison of LSs

Figure 14 depicts the area comparison between previous LSs and DIBLS. The measurements are conducted in terms of length and width (W/L) of the layout, and are expressed in square micrometers (µm^2^). It is notable that all previous LSs are designed with double cell height. The layout of the proposed DIBLS is presented in Figure 14. Despite being the smallest among the evaluated LS structures, the area of the proposed DIBLS remains competitive.

### 4.5. Layout Area of DIBLS

To validate the operational efficiency of the proposed DIBLS, an experimental test chip is optimized using a 45 nm technology node. The layout of the circuit, as depicted in Figure 15, serves as visual evidence of the completed LS with a multiplexer configuration.

The cell layout is created using Cadence Virtuoso, utilizing one metal and one double cell. In the DIBLS layout, the 2:1 MUX occupies the left side, while the LS is situated on the right side. Despite utilizing a total of eight transistors, the proposed DIBLS engages the minimum layout area of 5.84 µm^2^ among the available cells. The compact layout area of DIBLS can be endorsed to the minute sizes of most of the working transistors. Conversely, other available LSs are not area-efficient due to a variety of factors.

## 5. Conclusions

This article proposes a novel DIBLS, which is deployable in robotic arm controllers for leveraging the issue of distinct voltage level conversions, and uses a dynamic inverter designed to facilitate rapid and energy-efficient level conversions across an extensive voltage ranging from deep sub-threshold voltage levels to the supply voltage levels of cores. The integration of inverter-based LS components addresses issues related to current discrepancy and high static current. Additionally, the proposed DIBLS incorporates circuit optimization techniques aimed at further improving speed while simultaneously reducing power and energy consumption. DIBLS is optimized using a 45 nm CMOS technology, occupying a compact cell layout area of 5.84 µm^2^. DIBLS exhibits the capability to convert voltage levels from 0.3 V to 1.2 V and vice versa, at the frequency of 1 MHz. DIBLS demonstrates a delay of 0.22 ns, an energy per transition of 32.25 fJ, and a power consumption of 16.57 nW. Furthermore, the delay scalability of DIBLS is validated through experiments conducted under varying supply voltages. The promising performance characteristics make DIBLS a suitable candidate for applications ranging from the self-sustaining Internet of Things for robotic arm controlling with long-time standby modes to complex system-on-chip designs requiring ultra-wide-range voltage scaling capabilities

## Figures and Tables

**Figure 1 micromachines-15-01431-f001:**
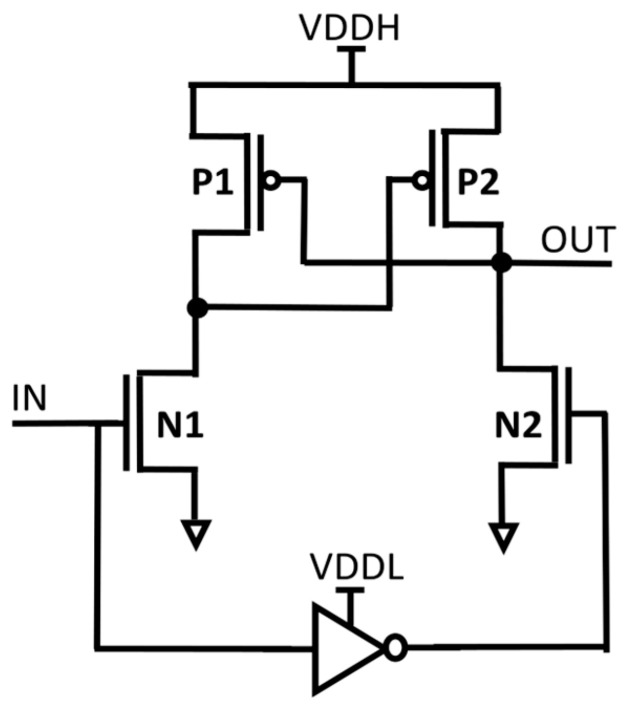
Structure of CPLS.

**Figure 2 micromachines-15-01431-f002:**
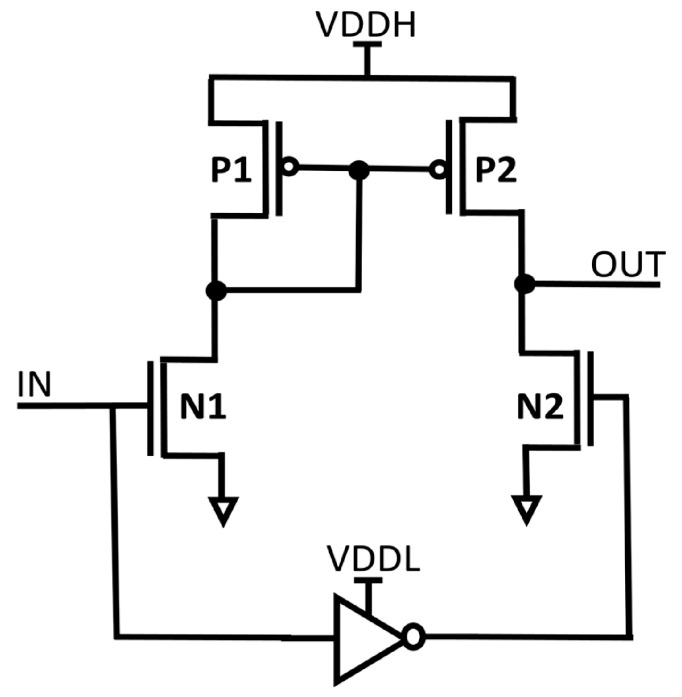
Structure of CMLS.

**Figure 3 micromachines-15-01431-f003:**
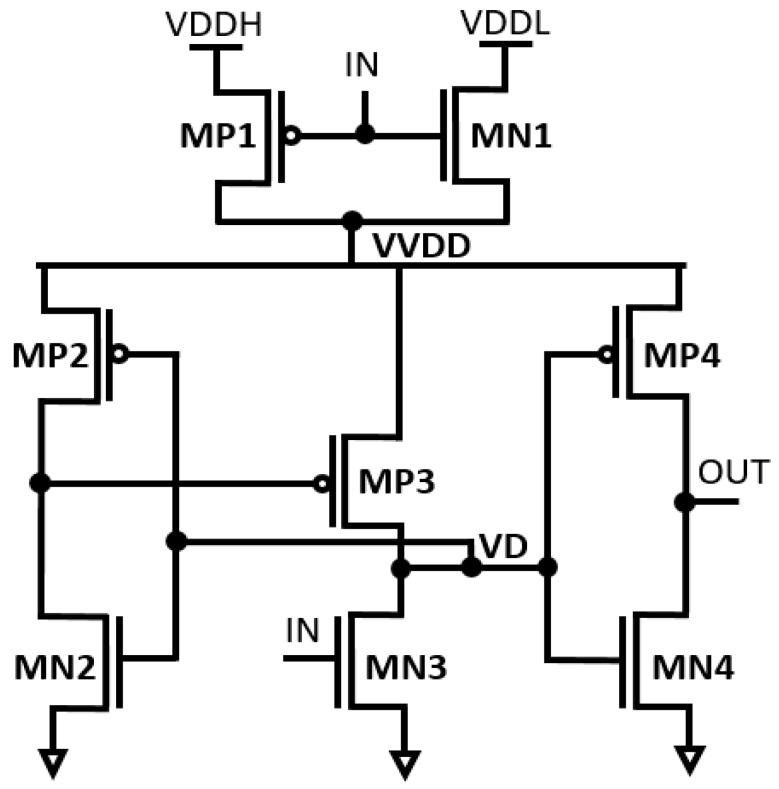
Structure of the proposed DIBLS.

**Figure 4 micromachines-15-01431-f004:**
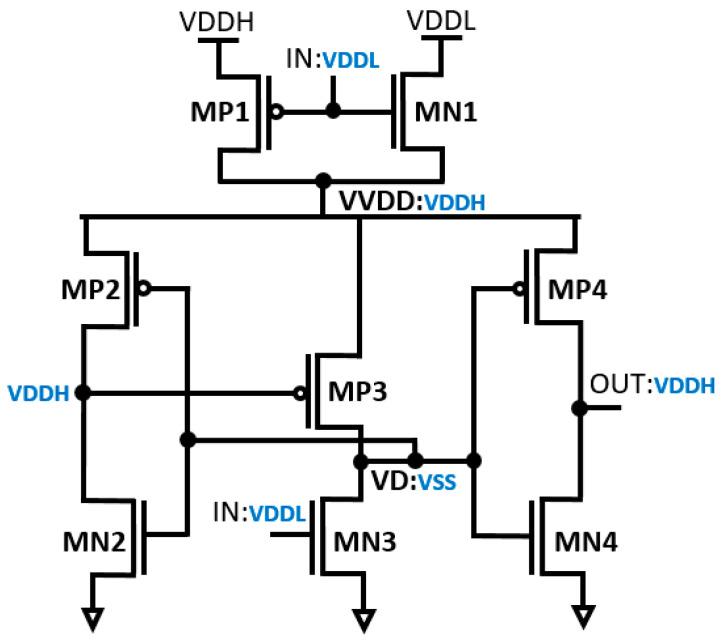
Transition from VDDL to VDDH.

**Figure 5 micromachines-15-01431-f005:**
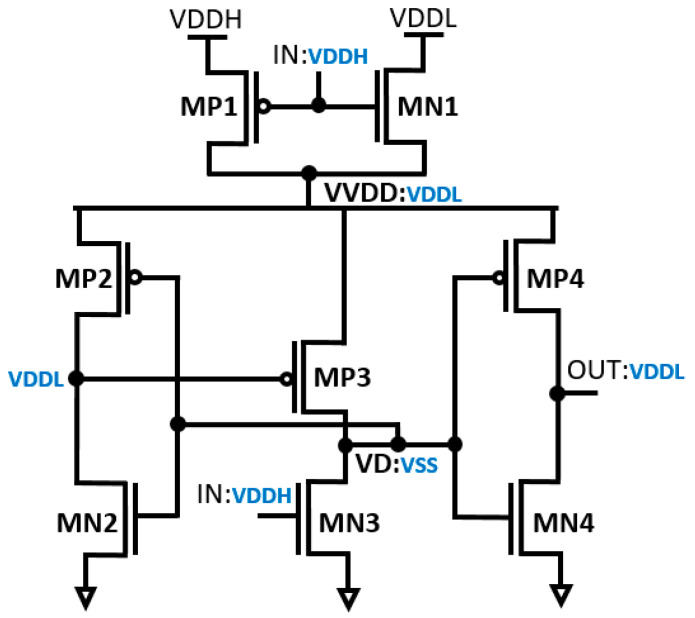
Transition from VDDH to VDDL.

**Figure 6 micromachines-15-01431-f006:**
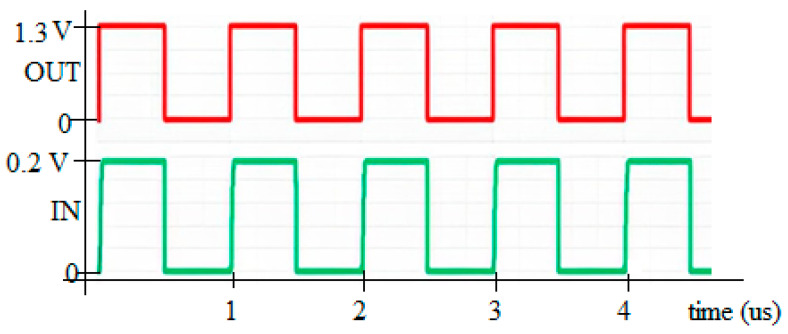
Response of DIBLS when VDDL = 0.3 V and VDDH = 1.2 V.

**Figure 7 micromachines-15-01431-f007:**
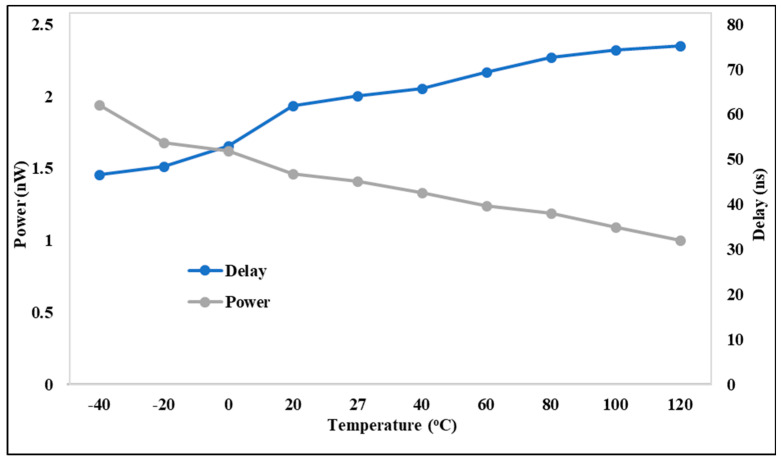
Behavior of DIBLS @ temperature.

**Figure 8 micromachines-15-01431-f008:**
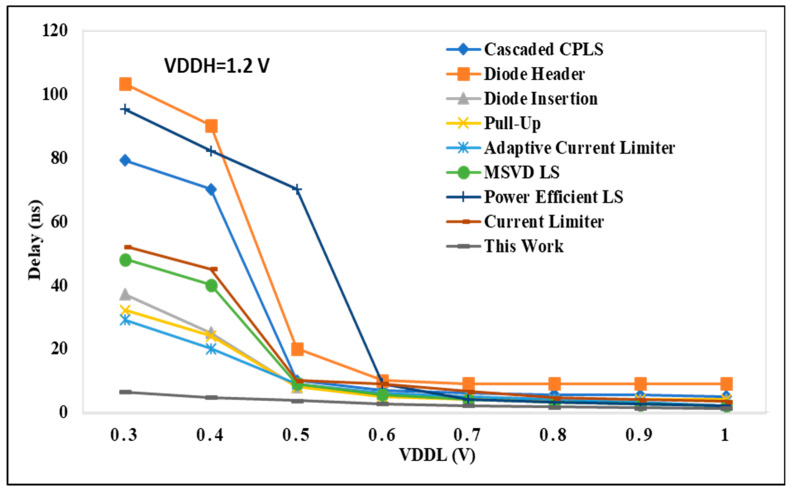
VDDL vs. delay @ VDDH = 1.20 V.

**Figure 9 micromachines-15-01431-f009:**
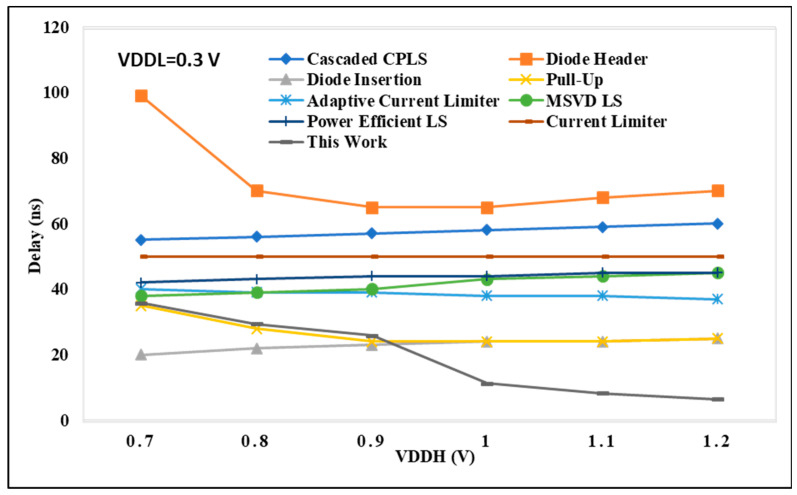
VDDH vs. delay@ VDDL = 0.30 V.

**Figure 10 micromachines-15-01431-f010:**
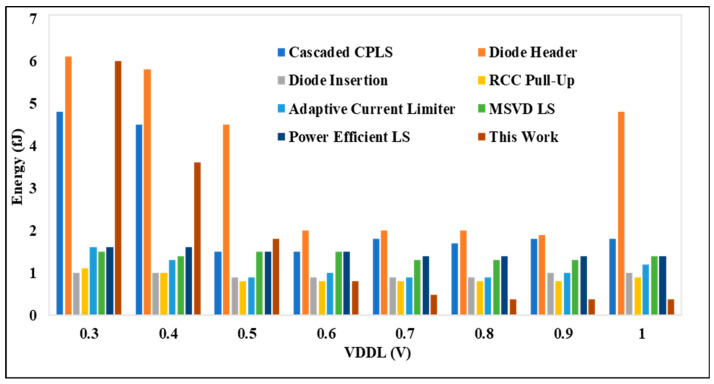
VDDL @ 0.3 V to 1 V vs. energy per transition.

**Figure 11 micromachines-15-01431-f011:**
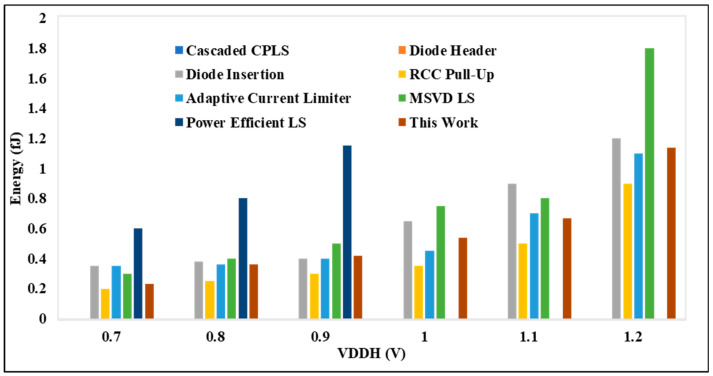
VDDH @ 0.7 V to 1.2 V vs. energy per transition.

**Figure 12 micromachines-15-01431-f012:**
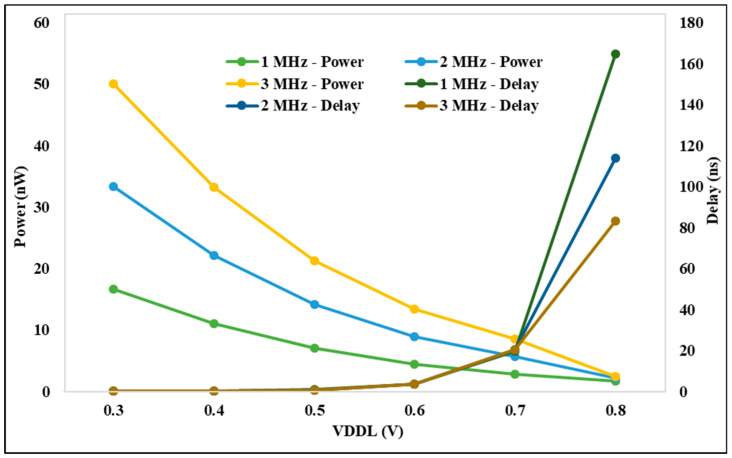
VDDL vs. power and delay @ different frequencies.

**Figure 13 micromachines-15-01431-f013:**
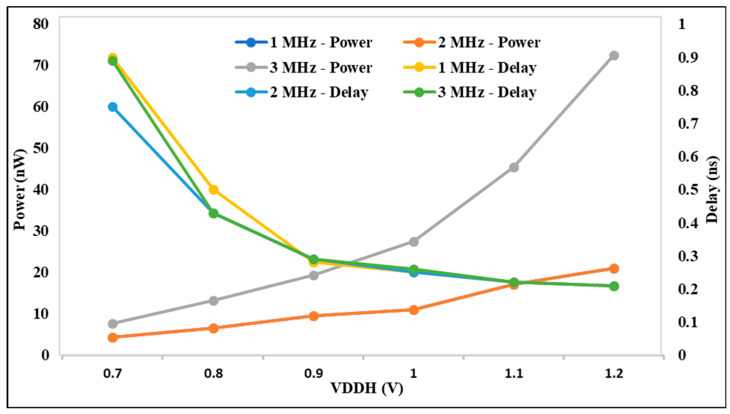
VDDH vs. power and delay @ different Frequencies.

**Figure 14 micromachines-15-01431-f014:**
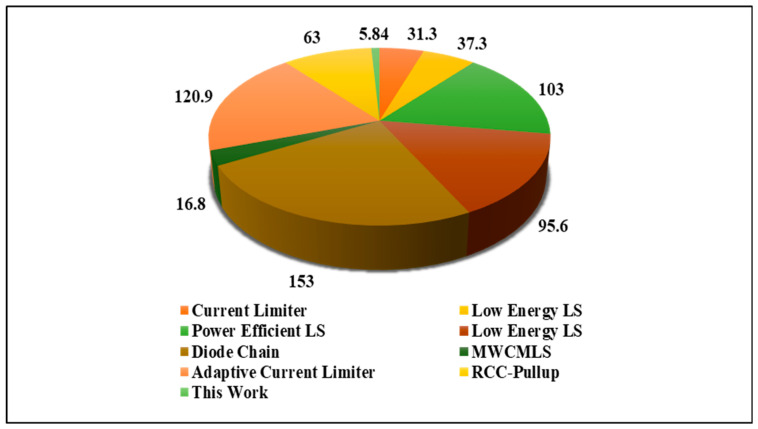
Area of proposed DIBLS and existing LSs.

**Figure 15 micromachines-15-01431-f015:**
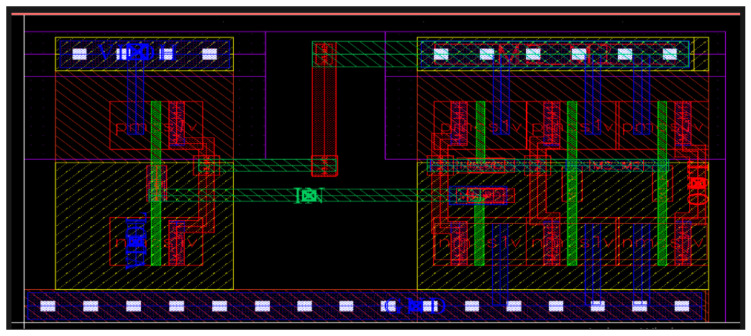
Layout of proposed circuit DIBLS.

**Table 1 micromachines-15-01431-t001:** Comparison of LSs.

Ref./ Proposed	Technology (nm)	Power (nW)	Delay (ns)	Area (µm^2^)
[21]	55	26.59	17.86	08.12
[22]	55	25.90	21.08	15.88
[23]	65	90.90	13.70	31.30
[24]	55	18.11	20.08	09.98
[25]	55	15.50	58.50	11.79
This work	45	16.57	0.220	05.84

## Data Availability

The original contributions presented in the study are included in the article, further inquiries can be directed to the corresponding author.

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
