# Peer review of "Design of Spurious Dynamic Inverter-Based Level Shifter with Error Tolerance for Robotic Arm Controller"

_micromachines, 2024, doi:10.3390/mi15121431_

Round 1
Reviewer 1 Report
Comments and Suggestions for Authors
In this paper, the author describes the development of a new type of signal level shifter (LS) intended for use in robotic arm controllers. The main focus is on the creation of a dynamic level shifter based on an inverter with an error correction circuit (Dynamic Inverter-Based Level Shifter, DIBLS). The practical significance of this work lies in the ability to effectively switch voltage levels between different domains, which is critical for intermodule interaction. One of the most significant achievements of this work (lines 150-300) is the solution to the problem of current disagreement that occurs when several components consume current together. The authors were able to overcome this obstacle by including feedback in the inverter (lines 250-270), which is controlled by the output node. This innovation ensures stabilization and stability of the level shifter, and also eliminates the threshold voltage drop, which has a positive effect on the operating speed of the device. The use of 45 nm CMOS technology in the creation of DIBLS allowed us to achieve impressive results. According to the data presented in the article, this level converter operates at voltages of 0.3 V and 1.2 V with a frequency of 1 MHz, while the power consumption is 16.57 nW, the delay is 0.22 ns, and the energy consumption per transition is 32.25 fJ. These figures indicate high energy efficiency and operating speed of the device, which is especially important for robotic systems, where the response speed plays an important role.
I would like to note that along with the positive aspects of the work, it has some drawbacks. The main one is the almost complete lack of analysis of the background of the study. The materials presented in lines 40-75 are insufficient. There are also a number of scientific comments. I will present them as a list.
Questions and comments of a scientific nature:
1. It is known that the performance of DIBLS varies significantly at different temperatures. However, the authors do not consider the stability of the work. Tell me, did you conduct this research? If yes, please mention it in the paper; if no, please mention the limitations.
2. When the gap between VDDL and VDDH is large, the speed and power consumption of DIBLS start to degrade, which reduces the efficiency under certain operating conditions. This may limit the use of this technology for scenarios that require stable operation under variable voltage conditions. I think it is critical to mention this in the paper in section 3 or 4.
3. I am not entirely clear about the architectural decisions that limit the current adjustment range and reduce flexibility under different operating conditions.
Errors of stylistic nature:
1. Line 24: "…based LSs employs current mirrors…" – error should be "employ"
2. Line 30, 52, 118, 130, 135, comma is missing. In line 166, I would add a comma before and, but this is at the discretion of the author.
3. Line 33: If I understand the context correctly, then "are demonstrating" should be replaced with "demonstrate" to match the tense.
4. In line 65. The text "…more efficient to decrease VDD near to Vth…" - "near to" I don't understand why the preposition "to" can be shortened to "near" for stylistic accuracy.
5. Line 194. I didn't understand anything at all. The text needs to be reformulated.
6. Figure 14. Not informative. I would delete it
7. Bibliography. Contains a huge number of errors. There are no doi, not all pages are indicated, publication dates are after the pages.
Conclusion. The article is generally positive. But the background of the study needs to be completely redone and the style of the text needs to be corrected.
Author Response
Comments and Suggestions for Authors
In this paper, the author describes the development of a new type of signal level shifter (LS) intended for use in robotic arm controllers. The main focus is on the creation of a dynamic level shifter based on an inverter with an error correction circuit (Dynamic Inverter-Based Level Shifter, DIBLS). The practical significance of this work lies in the ability to effectively switch voltage levels between different domains, which is critical for intermodule interaction. One of the most significant achievements of this work (lines 150-300) is the solution to the problem of current disagreement that occurs when several components consume current together. The authors were able to overcome this obstacle by including feedback in the inverter (lines 250-270), which is controlled by the output node. This innovation ensures stabilization and stability of the level shifter, and also eliminates the threshold voltage drop, which has a positive effect on the operating speed of the device. The use of 45 nm CMOS technology in the creation of DIBLS allowed us to achieve impressive results. According to the data presented in the article, this level converter operates at voltages of 0.3 V and 1.2 V with a frequency of 1 MHz, while the power consumption is 16.57 nW, the delay is 0.22 ns, and the energy consumption per transition is 32.25 fJ. These figures indicate high energy efficiency and operating speed of the device, which is especially important for robotic systems, where the response speed plays an important role.
Response: Thank you for positive comments
I would like to note that along with the positive aspects of the work, it has some drawbacks. The main one is the almost complete lack of analysis of the background of the study. The materials presented in lines 40-75 are insufficient. There are also a number of scientific comments. I will present them as a list.
Response: The description in the lines 40-75 is only introduction about the paper, demand for low power battery systems, different supply voltages, and conventional level shifters. In the literature survey (chapter 2) the complete state of art Level shifters and their background is described.
Questions and comments of a scientific nature:
- It is known that the performance of DIBLS varies significantly at different temperatures. However, the authors do not consider the stability of the work. Tell me, did you conduct this research? If yes, please mention it in the paper; if no, please mention the limitations.
Response: This research work was conducted in 45 nm technology node. The performance of the level shifter was assessed at different corners like temperature, variation in VDDL, variation in VDDH, Energy per transition, Power, Delay, Area, and Layout. As per my knowledge and even level shifter papers from the references have not described about stability of the work.
- When the gap between VDDL and VDDH is large, the speed and power consumption of DIBLS start to degrade, which reduces the efficiency under certain operating conditions. This may limit the use of this technology for scenarios that require stable operation under variable voltage conditions. I think it is critical to mention this in the paper in section 3 or 4.
Response: Quite naturally the efficiency decrease at extreme operation conditions. In real time scenarios nominal VDDL and VDDHs are enough, in that case the speed and power consumption are good. When it comes to extreme scenarios there is degradation in speed and power consumption.
- I am not entirely clear about the architectural decisions that limit the current adjustment range and reduce flexibility under different operating conditions.
Response: It is clearly described in the paper in chapter 3.1 and 3.2
Errors of stylistic nature:
- Line 24: "…based LSs employs current mirrors…" – error should be "employ"
Response: Corrected in the revised paper
- Line 30, 52, 118, 130, 135, comma is missing. In line 166, I would add a comma before and, but this is at the discretion of the author.
Response: Corrected in the revised paper
- Line 33: If I understand the context correctly, then "are demonstrating" should be replaced with "demonstrate" to match the tense.
Response: Corrected in the revised paper
- In line 65. The text "…more efficient to decrease VDD near to Vth…" - "near to" I don't understand why the preposition "to" can be shortened to "near" for stylistic accuracy.
Response: Corrected in the revised paper
- Line 194. I didn't understand anything at all. The text needs to be reformulated.
Response: Reformulated the sentence in the revised paper
- Figure 14. Not informative. I would delete it
Response: Area of the level shifters in micrometers, in some sense it will be useful
- Bibliography. Contains a huge number of errors. There are no doi, not all pages are indicated, publication dates are after the pages.
Response: Complete concentration laid on the quality of work and results analysis. Doi and other related bibliography will be appended in the proof reading stage. Honorable editorial team will support in this regard.
Conclusion. The article is generally positive. But the background of the study needs to be completely redone and the style of the text needs to be corrected.
Response: Thank you for positive comments, the background of the study like introduction, literature study, results and discussions of the proposed work is reviewed and corrected possible extent and changed the text to red color for easily identification.

Reviewer 2 Report
Comments and Suggestions for Authors
It is an interesting paper. I am unable to point out any substantive shortcomings in the manuscript. I have two comments that could contribute to improving the manuscript, i.e.
1. In the second chapter, the competitive solutions described for level-shifter systems can be presented as schematics similar to those in Figure 1, 2 and 3. There are more technologies of level-shifters mentioned than only two given in Fig. 1 and Fig.2.
2. Section 4 lacks a photo of the research setup on which the measurement was performed using the least developed DIBLS system (the others are unnecessary).
Thank you.
Comments on the Quality of English LanguageIn the manuscript there are lots of grammatical errors (examples are below):
L24: ...LSs employ(s is wrong)
L36: ...entire process (missing is) executed in 45 nm..
L92: there is: at the voltage of 360 mV and an frequency but should be: at a voltage of 360 mV and a frequency ...
L121: there is: "These LS demonstrates..." but should be "This LS demonstrates..."
L123: there is: "well lower the threshold voltage...", then is missing
Similar errors are found in the whole manuscript.
Author Response
We would like to thank the esteemed reviewers for their valuable suggestions. We appreciate your comments and have carefully considered your suggestion
Answers for Comments and Suggestions from Reviewers:
Title: Design of Spurious Dynamic Inverter-Based Level Shifter with Error tolerance for Robotic Arm controller
Reviewer 2 Round 1
Comments and Suggestions for Authors
It is an interesting paper. I am unable to point out any substantive shortcomings in the manuscript. I have two comments that could contribute to improving the manuscript, i.e.
Response: Thank you for positive comments
- In the second chapter, the competitive solutions described for level-shifter systems can be presented as schematics similar to those in Figure 1, 2 and 3. There are more technologies of level-shifters mentioned than only two given in Fig. 1 and Fig.2.
Response: The conventional level shifters are described in introduction chapter in figure 1and 2. The start of art level shifters are described in literature survey chapter describes about 15 types of level shifters. Describing 15 level shifters with 15 schematics will become a huge page occupying and licensing issues. In this aspect reader can refer the references for further knowing about schematics.
- Section 4 lacks a photo of the research setup on which the measurement was performed using the least developed DIBLS system (the others are unnecessary).
Response: Section 4 describes about performance analysis of the level shifters in terms of LS Delay, analysis of Level Shifter Energy, Level Shifter Power Delay product, Area comparison, and Layout area of DIBLS.
Comments on the Quality of English Language
L24: ...LSs employ(s is wrong)
Response: Corrected in the revised paper
L36: ...entire process (missing is) executed in 45 nm..
Response: Corrected in the revised paper
L92: there is: at the voltage of 360 mV and an frequency but should be: at a voltage of 360 mV and a frequency ...
Response: Corrected in the revised paper
L121: there is: "These LS demonstrates..." but should be "This LS demonstrates..."
Response: Corrected in the revised paper
L123: there is: "well lower the threshold voltage...", then is missing
Response: Corrected in the revised paper
Similar errors are found in the whole manuscript.
Response: Possible extent corrected in the revised paper

Round 2
Reviewer 1 Report
Comments and Suggestions for Authors
all comments have been resolved